Effect of sit-to-stand-based training on muscle quality in sedentary adults: a randomized controlled trial

Lizama-Pérez Rodrigo 1 2
http://orcid.org/0000-0002-1008-176X Chirosa-Ríos Luis Javier 1
Contreras-Díaz Guido 3
http://orcid.org/0000-0002-6878-8004 Jerez-Mayorga Daniel 1 4 daniel.jerez@unab.cl
Jiménez-Lupión Daniel 1
http://orcid.org/0000-0001-9006-8516 Chirosa-Ríos Ignacio Jesús 1
1 Department of Physical Education and Sports, University of Granada , Granada , Spain
2 Departamento de Ciencias Morfológicas, Facultad de Medicina y Ciencia, Universidad San Sebastián , Valdivia , Chile
3 Department of Health, University of Los Lagos , Puerto Montt, Llanquihue , Chile
4 Exercise and Rehabilitation Sciences Institute, School of Physical Therapy, Faculty of Rehabilitation Sciences, Universidad Andres Bello , Santiago , Chile
Nuhmani Shibili
Electronic publication date: 2023 Jul 12
Publication date: 2023
Volume: 11
Electronic Location ID: e15665
Received 2023 Apr 19; Accepted 2023 Jun 8
Copyright: © 2023 Lizama-Pérez et al.
Copyright year: 2023
Copyright holder: Lizama-Pérez et al.
License: This is an open access article distributed under the terms of the Creative Commons Attribution License, which permits unrestricted use, distribution, reproduction and adaptation in any medium and for any purpose provided that it is properly attributed. For attribution, the original author(s), title, publication source (PeerJ) and either DOI or URL of the article must be cited.
License URL: https://creativecommons.org/licenses/by/4.0/

Keywords: Muscle power, Sarcopenia, Muscle quality index, Chair stand, Muscle architecture

Funding: Recualificación del Profesorado Universitario. Modalidad Margarita Salas Universidad de Granada/Ministerio de Universidades y Fondos Next Generation de la Unión Europea DGI-University Andres Bello No. DI-6-20/CBC Daniel Jerez-Mayorga has a contract through the program “Recualificación del Profesorado Universitario. Modalidad Margarita Salas”, Universidad de Granada/Ministerio de Universidades y Fondos Next Generation de la Unión Europea. The DGI-University Andres Bello, N° DI-6-20/CBC supported the publication fee for this article. The funders had no role in study design, data collection and analysis, decision to publish, or preparation of the manuscript.

==============================
The aim of this study was to compare the effects of sit-to-stand (STS) training programs with 5 vs. 10 repetitions on muscle architecture and muscle function in sedentary adults. Sixty participants were randomly assigned into three groups: five-repetition STS (5STS), 10-repetition STS (10STS), or a control group (CG). Participants performed three sets of five or 10 repetitions of the STS exercise three times per week for 8 weeks. Before and after 8 weeks, all groups performed ultrasound measures to evaluate muscle thickness (MT), pennation angle (PA), and fascicle length (FL), and the five-repetition STS test to estimate the relative STS power and muscle quality index (MQI). After 8 weeks, both experimental groups improved MQI (40–45%), relative STS power (29–38%), and MT (8–9%) (all p < 0.001; no differences between the 5STS vs. 10STS groups). These improvements in both groups resulted in differences regarding the CG, which did not present any change. In addition, only the 5STS group improved PA (15%; p = 0.008) without differences to the 10STS and CG.This suggests that STS training is time-effective and low-cost for improving muscle function and generating adaptations in muscle architecture.

Introduction

Muscle power is a key factor for the health and well-being of individuals, as it not only allows them to carry out daily physical activities (Alcazar et al., 2020) but also plays a crucial role in the prevention of chronic diseases and in increasing the quality of life as we age (Reid & Fielding, 2012). Loss of muscle power, which can occur independently of muscle mass loss, has serious consequences such as decreased mobility, independence, risk of falls and, quality of life (Alcazar et al., 2023; Jiménez-Lupión et al., 2023). Conversely, maintaining optimal levels of muscle power can improve athletic abilities, reduce the risk of injury (Suchomel, Nimphius & Stone, 2016), and be associated with reduced healthcare expenditures and mortality (Tieland, Trouwborst & Clark, 2018).

Muscle power is defined as the product of muscle contraction force and velocity, and its development involves both morphological and neural factors (Suchomel et al., 2018). Therefore, it is necessary to consider both the functional and structural parameters for evaluation (Fragala, Kenny & Kuchel, 2015). In this scenario, knowing the muscle architecture would provide a more complete view of the muscle function (Narici et al., 2021). Muscle architecture is defined as the arrangement of muscle fibers within a muscle in relation to the axis of force generation (Lieber & Fridén, 2000), and is one of the most important components in its function (Lieber & Ward, 2011). Recent studies have determined the relationship between muscle architectural parameters and functional and health components in humans (Coratella, Rinaldo & Schena, 2018).

The muscle quality index (MQI), which is computed using the sit-to-stand test (STS) time and a formula that accounts for anthropometric variables, body mass, and gravity, has recently been utilized as a way to evaluate muscular power (Brown, Harhay & Harhay, 2016; Takai et al., 2009). STS has been described as an essential clinical test to assess functional capacity and quality of life in the older adults population, as it is cost-effective, time- and space-efficient, and easy to administer (Alcazar et al., 2018; Jerez-mayorga et al., 2020). In addition, it has been used as a therapeutic agent in different populations and has shown favorable results (Chaovalit, Taylor & Dodd, 2020; Zampogna et al., 2021).

It is widely acknowledged that incorporating exercise into daily life is essential for maintaining good health (Hegde, 2018), but many people have difficulties due to a lack of time and motivation (Blake, Stanulewicz & McGill, 2017; Portela-Pino, Valverde-Esteve & Martínez-Patiño, 2021). Therefore, it is important to create efficient and low-cost routines that require little time to increase adherence, especially in older adults (Nicolson et al., 2017). This represents a challenge for health and sports professionals who must find ways to motivate patients to follow exercise programs to achieve positive results (Spiteri et al., 2019).

An alternative training option for this situation is to use STS as a training method, where an improvement in performance, balance, and strength has been demonstrated after STS interventions (Chaovalit, Taylor & Dodd, 2020; Hyun, Lee & Lee, 2021). In this context, the STS task could effectively assess and improve muscle power in sedentary individuals, as it requires strength and speed to perform movements. However, there is a paucity of studies using this test as an intervention method in sedentary individuals, especially if we intend to evaluate its effect on function and muscle architecture. Therefore, the aim of this study was to compare the effects of sit-to-stand (STS) training programs with 5 vs. 10 repetitions on muscle architecture and muscle function in sedentary adults.

We hypothesized that STS training would increase muscle function and produce adaptations in the architecture of the vastus lateralis muscle (VL) in sedentary adults.

Materials and Methods

Study design

This was a three-arm, parallel-group, randomized trial. The study was conducted at the Basic Sciences Laboratory, Universidad San Sebastian, Valdivia, Chile. Recruitment was conducted through social media and email during the months of June, July, and August 2022. The study protocol was approved by the Scientific Ethics Committee of the University of Granada, Spain (2380/CEIH/2021) and of the Universidad San Sebastián, Chile (55-2021-20). It has also been registered in the Protocol Registration and Results System (https://clinicaltrials.gov/ct2/show/NCT05469191).

Allocation

The principal investigator generated a randomization sequence using an Stata 17.0 with blocks of varying sizes that were combined randomly. The allocation was performed in a one-to-one ratio, and allocation concealment was ensured by using sealed envelopes.

Blinding

Neither the participants nor the providers were blinded in this study.

All participants were invited to a familiarization visit in the laboratory. During this visit, the descriptive, biometric, and medical variables were assessed for each participant. We also determined whether or not the participants met the inclusion criteria and did not meet the exclusion criteria. In addition, VL muscle architecture of the dominant lower limb was assessed.

Synchronous and asynchronous work sessions were conducted. All training sessions, both synchronous and asynchronous, were meticulously designed and supervised. The synchronous sessions were directly supervised by the lead investigator, ensuring that rest times and training intensity were consistently controlled. For the asynchronous sessions, participants were previously familiarized with the components of the training, enabling them to manage rest times, maintain appropriate intensity levels, and ensure safety during their workout routines. To verify that the asynchronous sessions were being correctly performed by the participants, they were asked to submit a video of the session. On average, both types of sessions had a duration of 2.7 ± 2.4 min.

Rest times were monitored to ensure the protocol was executed correctly, and a Borg CR-10 scale was used to assess perceived exertion (Shariat et al., 2018). As part of the safety protocol, a perceived exertion of 6 was set as the maximum limit for each set. If this value was exceeded, participants were instructed to stop the routine. The scale was explained to each participant, and they were familiarized with perceived exertion before the intervention began. A new assessment of the variables was performed 1 week after completing the protocol (Fig. 1). All the participants completed 100% of the training sessions.

Figure 1 Design of the investigation.

Participants

Statistical software (G*Power, v3.1.9.7; Heinrich-Heine-Universität, Düsseldorf, Germany) was used to calculate the sample size. A moderate effect size of 0.95, obtained from a previous study (Ribeiro et al., 2020) was used. Considering the above and a desired power (error 1−ß) = 0.95, alpha error <0.05, the total sample size was 18 participants per group. However, in anticipation of possible dropouts from the study, 20 participants per group were included.

The participants selected for this study met the following criteria: 18 years of age or older, sedentary, and able to perform STS independently. On the other hand, participants were excluded if they were professionally involved in sports or resistance training, had untreated hypertension, acute neuromuscular or joint injury, or had suffered an acute myocardial infarction or fracture in the last six months. Informed consent forms detailing the study information, which followed the principles of the Declaration of Helsinki, were obtained written from all participants prior to the start of the interventions.

Primary outcomes

Muscle function

The Muscle Quality Index (MQI) was calculated using the formula proposed by Fragala, Kenny & Kuchel (2015). This formula employs the time required for five sit-to-stand (STS) repetitions: MQI = ((leg length × chair height) × body mass × gravity × 10/STS Time). It takes into consideration the leg length in meters, the height of the chair used in the test, body mass in kilograms, acceleration due to gravity (9.81 m/s2), and a constant of 10.

Additionally, the relative mean STS power (W·Kg-1) was calculated using the results from the five-times sit-to-stand (5STS) test, as proposed by Alcazar et al. (2018). The formula used for this calculation is as follows:

Relative mean STS power = 0.9 × g × Height (0.5 × Chair height)/(Five STS time × 0.1).

Secondary outcomes

Ultrasonography

In this study, ultrasound images were used to evaluate the muscle thickness (MT), pennation angle (PA), and fascicle length (FL) in each participant’s dominant lower limb. The probe was positioned on the longitudinal axis of the VL, and a linear B-mode probe with a frequency range of 7.5 10 MHz was used at a depth of 8 cm (Sonus, DUO LCP). The probe was coated with a water-soluble gel to prevent skin pressure. Participants were seated with their knees flexed at 90° and instructed to relax their muscles at the time of measurement and avoid exercise for 48 h before the assessment. Measurements were taken at 50% of the distance between the greater trochanter and lateral condyle of the femur, and three images were captured at each site during the evaluation sessions. ImageJ software (ImageJ 1.42; National Institutes of Health, Bethesda, MD, USA,) was used to process and analyze the images, and MT was measured as the average distance between the superficial and deep aponeuroses. PA was measured as the angle between the fascicle and deep aponeurosis, and FL was calculated using the following formula: FL= sin(y+90°) × MT/sin(180°−(y+180°−PA)), where y is the angle between the superficial and deep aponeurosis, and AP is the pennation angle (Blazevich, Gill & Zhou, 2006; Perkisas et al., 2021). All measurements and image analyses were performed by the same investigator.

Anthropometry

Body composition was assessed using bioelectrical impedance tetrapolar analysis (Rice Lake Body Composition D1000-3; Rice Lake, WI, USA). The stature was measured using a portable stadiometer (SECA, Model 213, Hamburg, Germany to 0.1 cm). Leg length was measured manually using an anthropometric measurement protocol (Norton, 2019). The leg length was defined as the distance (in meters) from the greater trochanter of the femur to the lateral malleolus (Marzilger et al., 2020).

Sit-to-stand test

The time required for participants to complete five repetitions of the stand-and-sit exercise was established. During the evaluation, participants were advised to perform the repetitions as swiftly as possible, with their arms crossed over their chests at shoulder level. Three sets of five repetitions were timed, with a 1-min rest interval between each set. The quickest time recorded was selected for further analysis (Alcazar et al., 2018).

Sit-to-stand training program

During the first 4 weeks, the 5STS and 10STS groups performed the STS exercise three times per week. The 5STS performed three sets of five repetitions, and the 10STS three sets of 10 repetitions. A rest time of 30/60 s was considered for each set. During the last 4 weeks, the number of sets increased to five to continue progressing in the training. Importantly, participants were asked to perform each set of STS movements as fast as possible and against the clock. The rest times were monitored to ensure that the protocol was performed correctly, and a Borg CR-10 scale was used to assess perceived exertion (Shariat et al., 2018).

Statistical analyses

Descriptive data are presented as mean ± standard deviation, with normality verified using the Shapiro-Wilk normality test. Additionally, the intraclass correlation coefficient (ICC), the standard error of measurement (SEM), and the minimal detectable change (MDC) were calculated (Sainani, 2017). Subsequently, a 2 (time) × 3 (group) repeated measures ANOVA with a Bonferroni post hoc test was employed to examine differences between time points and groups. Effect size was interpreted using Cohen’s d scale, in which values less than 0.20 are considered trivial, between 0.20–0.59 are small, 0.60–1.19 are moderate, 1.20–2.00 are large, and values greater than 2.00 are considered very large. All analyses were conducted using JASP software (version 0.17). The threshold for statistical significance was set at p ≤ 0.05.

Results

Flow of participants and sample characteristics

Figure 2 shows a flowchart of the participant. Of the 64 participants evaluated, 60 met the eligibility criteria. During follow-up, one participant from group five STS dropped out of the study due to personal health problems and at the end of the study, one participant from the control group was excluded from the analysis for not attending the post-evaluation due to time constraints. The characteristics of the participants are summarized in Table 1.

Figure 2 CONSORT diagram of eligibility and inclusion of participants.

Table 1 Characteristics of the participants.

Group	Sex	Age (years)	Height (cm)	Weight (kg)	BMI (kg/m²)	Leg length (m)	Leg muscle mass (kg)	
5STS	11 females/9 males	42 ± 3.6	162 ± 9.0	80 ± 3.2	30 ± 5.2	0.8 ± 0.1	8.7 ± 1.8	
10STS	15 females/4 males	42 ± 9.7	162 ± 7.6	75 ± 14.3	28 ± 4.1	0.8 ± 0.1	9.3 ± 1.9	
CG	9 females/10 males	33 ± 12.0	167 ± 9.3	80 ± 16.3	28 ± 5	0.8 ± 0.04	9.9 ± 2.1	
Note:

Values are shown as mean ± standard deviation.

Muscle function

The study revealed significant time and group interactions for the MQI and relative STS power after 8 weeks of intervention. Specifically, both the 5STS and 10STS groups showed significant improvements in these parameters, with no notable differences between them.

When comparing the pre- and post-measurements for each group, all groups demonstrated an improvement in MQI. However, the improvements were more pronounced and significant in the 5STS and 10STS groups compared to the control group.

In terms of relative STS mean power, both the 5STS and 10STS groups exhibited a significant increase at the end of the study period, while the control group did not show significant changes (Table 2). The individual changes for each variable are graphically represented in Fig. 3.

Table 2 Changes in muscle function.

	Group	PRE
mean ± SD	POST
mean ± SD	p-value	Percentage change (%)	95% CI for mean difference	Time x Group interaction	
Mean difference	Lower	Upper	F	p-value	Effect size	
MQI (W)
Fragala et al.	Control	579.5 ± 152.5	606.1 ± 163.3	1.0	+5	−150.568	−171.114	−130.021	36.452	<0.001	0.201	Small	
5STS	502.9 ± 170.5	706.5 ± 157.8 ***	<0.001	+40	
10STS	488.5 ± 148.2	710.1 ± 126.6 ***	<0.001	+45	
Relative STS mean power (W·Kg-1)	Control	7.4 ± 0.9	7.5 ± 1.2	1.0	+1	−1.451	−1.641	−1.262	53.290	<0.001	0.129	Trivial	
5STS	6.5 ± 1.6	8.4 ± 1.3 ***	<0.001	+29	
10STS	6.4 ± 1.2	8.8 ± 1.19***#	<0.001	+38	
Notes:

MQI, muscle quality index.

*** = Significant change p-value < 0.001 from pre-evaluation.

# = Significant change p-value < 0.05 vs. Control.

Figure 3 Individual changes in functional performance.

Muscle architecture

At the beginning of the study, no significant differences were detected among the 5STS, 10STS, and CG in any of the muscle architecture variables. Following the intervention, significant time × group interactions were observed for both MT and PA, but not for FL.

When comparing the pre- and post-intervention measurements in each group, significant increases in MT were detected in both the 5STS and 10STS groups. Similarly, only the 5STS group showed a significant increase in PA. Conversely, in the 10STS group, while an increase in PA was observed, it was not significant. In all groups, a slight decrease in FL was noted, being more pronounced in the intervention groups, although these changes were not significant.

As for the CG, no significant changes were recorded in any of the studied muscle architecture variables: MT, PA, and FL (Table 3). Individual differences in muscle architecture are shown in Fig. 4.

Table 3 Changes in muscle architecture of VL.

	Group	PRE
mean ± SD	POST
mean ± SD	p-value	% Change	95% CI for mean difference	Time x Group interaction	
	Mean difference	Lower	Upper	F	p-value	Effect size	
MT (cm)	Control	2.6 ± 0.54	2.6 ± 0.51	0.307	−5%	−0.110	−0.164	−0.055	16.978	<0.001	0.21	Small	
5STS	2.54 ± 0.47	2.78 ± 0.5 ***	<0.001	+9%	
10STS	2.49 (0.56)	2.70 ± 0.53 **	0.001	+8%	
FL (cm)	Control	12.35 ± 3.21	12.1 ± 2.39	–	−2%	0.669	−0.140	1.479	0.357	0.701	0.003	Trivial	
5STS	12.83 ± 3.21	11.72 ± 2.29	–	−9%	
10STS	12.25 ± 4.5	11.63 ± 2.11	–	−5%	
PA (°)	Control	12.08 ± 2.46	11.87 ± 2.05	1.0	−2%	−1.025	−1.603	−0.448	4.646	0.014	0.025	Small	
5STS	12.09 ± 2.45	13.9 ± 2.23 **	0.008	+15%	
10STS	12.09 ± 2.08	13.56 ± 2.13	0.073	+13%	
Notes:

MT, muscle thickness; PA, pennation angle. FL, fascicle length; VL, vastus lateralis.

** = Significant change p-value < 0.01 from pre-evaluation.

*** = Significant change p-value < 0.001 from pre-evaluation.

Figure 4 Individual changes in VL muscle architecture.

MT, muscle thickness; PA, pennation angle; FL, fascicle length.

Discussion

The objective of this research was to compare the effects of five and 10 repetitions of STS training programs on muscle function and muscle architecture in sedentary adults. The main findings of the study were that both the five and 10 repetitions of STS programs over an 8-week period resulted in significant increases in the MQI and relative STS mean power. Additionally, adaptations in muscle architecture parameters were observed, including an increase in MT and PA. However, the increase in PA was significant only in the 5STS group. These findings suggest that STS, as a low-cost exercise, can improve the muscle function of sedentary individuals and promote adaptations in muscle architecture. It is important to note that this training program is also characterized by its time efficiency. These findings support the implementation of STS programs as a viable and efficient strategy to improve function and muscle architecture in sedentary adults.

There are several methods to evaluate muscle quality (Fragala, Kenny & Kuchel, 2015; Correa-de-araujo et al., 2017); however, many are normalized to muscle mass (Naimo et al., 2021), which remains an important factor for evaluation. The instruments required for these evaluations are often difficult to obtain or incur high costs (Ticinesi et al., 2016). In this context, MQI through STS is a useful, low-cost, and rapid tool for measuring muscle quality in patients in a clinical context (Brown, Harhay & Harhay, 2016; Jerez-mayorga et al., 2020). Its significant increase could indicate an improvement in the muscle quality and functional status of the lower extremities of the participants (Scanlon et al., 2014), without considering other parameters, such as the neural component (Fragala, Kenny & Kuchel, 2015). Similarly, the values provided by the Alcazar et al. (2018) formula for muscle velocity and power provide an important indicator for measuring the muscular function of the lower extremities and thus for monitoring our patients.

There was a significant increase in PA (5STS) and a slight, non-significant decrease in the FL intervention group. Previous studies found an increase in PA in response to concentric contraction training (Franchi et al., 2017) and a reduction in FL in response to concentric exercise interventions (Timmins et al., 2016), which is opposite to what occurs in eccentric interventions (Alonso-Fernandez et al., 2020). However, all of these studies used more demanding exercise doses and higher volumes (Lizama-Pérez et al., 2022). The increase in PA and slight decrease in FL in this study could be explained by the type of exercise used, as we considered STS to behave as a concentric loading exercise for the VL because of the need to overcome gravity during knee extension, acting mostly with a concentric component (Frykberg & Ha, 2015; Franchi et al., 2017; Lizama-Pérez et al., 2022). PA is considered an essential factor in force production (Lieber & Ward, 2011; Narici, Franchi & Maganaris, 2016), possibly by allowing a greater number of contractile fibers in the aponeurosis area (Blazevich & Sharp, 2005). Although the association of PA with force production has recently been questioned (Lieber, 2022), its behavior in response to different interventions is still not clear, possibly because other variables need to be considered in this discussion, such as the role of non-contractile tissue and dynamic studies that appreciate the rotation of the fascicle during contraction (Son, Rymer & Lee, 2020; Lieber, 2022).

Another parameter evaluated in this intervention was the adaptation of the VL in the MT, which increased significantly in both the intervention groups. This situation has been observed in response to different types of interventions. MT has been described as a valuable predictor of muscle strength during knee extension (Abe, Loenneke & Thiebaud, 2015), although further studies are needed to use it as a predictor of muscle function (Casey et al., 2022). Similarly, it has been proposed as a promising tool for predicting sarcopenia in older adults (Ticinesi et al., 2016; Casey et al., 2022). Therefore, given the limited availability of instruments for evaluating muscle mass using bioimpedance and dual-energy X-ray absorptiometry (DXA), or physiological cross-sectional area through magnetic resonance imaging (Perkisas et al., 2021), MT measured using ultrasound can be a useful indicator in the evaluation of patients.

In recent years, various interventions have been tested to examine their effects on health outcomes from low-energy cost routines. Data suggests that low-volume, high-load training can enhance strength (Fisher et al., 2014). However, it appears that exercises with light to moderate loads, executed at the highest possible speed, are more effective in improving functional capacity in older adults (Ramírez-Campillo et al., 2014; Balachandran et al., 2022). A recent meta-analysis indicates that while multiple sets per exercise can slightly increase lower limb strength in middle-aged and older adults, a single set per exercise is sufficient for enhancing upper limb strength, muscle size, and functional capacity in these age groups (Marques et al., 2023). Lastly, the observed deficits in the rate of force development (RFD) in older adults underscore the importance of muscle power in fall prevention. Older adults 10d to exhibit greater deficiencies in RFD compared to maximal strength, suggesting that power training could be particularly beneficial for this demographic (Jiménez-Lupión et al., 2023).

Exercise routines have been tested that interrupt sedentary time and improve insulin and glucose levels. (Dempsey et al., 2016a, 2016b). Other studies have used STS as an intervention to evaluate its effects on functional variables. Hyun, Lee & Lee (2021) evaluated its effect after 6 weeks of intervention in stroke patients, who significantly increased their strength, balance, and speed. Similarly, Matsufuji et al. (2015) applied an STS intervention to elderly hemodialysis patients, who significantly improved their functional independence. However, these studies were conducted in a population different from that used in this study, and with different STS modalities not applied to improve muscular power. Although other STS modalities exist (Bohannon & Crouch, 2019; Figueiredo et al., 2021), We use those that take into account the time taken to complete the five repetitions, rather than the number of repetitions performed, as a measure of the result. By trying to beat the time, we believe that this modality encourages faster movements that contribute to improving lower limb muscle power.

A limitation of the present study was the heterogeneity in the age of the population. Additionally, to complement the effectiveness of the intervention, additional measurements, such as dynamometric devices to assess strength and velocity, are needed. Another limitation of the study is the absence of an evaluator during the training sessions, which resulted in participants not receiving real-time feedback while performing the exercises to encourage them to complete each repetition at maximum speed. Furthermore, despite being a controlled trial, it was not blinded, as both participants and evaluators were aware of the study conditions and their assigned groups. This lack of blinding may have introduced biases in the data collection and analysis, as expectations and preconceptions could have influenced the results and their interpretation. Additionally, it is worth noting that the measurements of muscular architecture and subsequent analysis were conducted by the same evaluator. Therefore, caution is necessary when generalizing the findings of this study, and these limitations should be considered when interpreting the results.

Conclusions

In conclusion, the results of this study suggest that STS training effectively improves muscle quality and generates adaptations in the AP and MT muscles. These findings are important to understand that although a low volume of training was used, the results were positive in the study population. This finding indicates the importance of encouraging the general population to engage in exercise, even with a seemingly low volume. However, further research is required to determine the ideal training volume. Additionally, it is essential to study its effects on functional variables that support improvement in daily activities and promote independence in the elderly population.

Supplemental Information

Supplemental Information 1 Reporting checklist for randomised trial.

Click here for additional data file.

Supplemental Information 2 Categorical Data.

Click here for additional data file.

Supplemental Information 3 Study Data.

Pre and post intervention variables.

Click here for additional data file.

This article will be part of the Rodrigo Lizama-Pérez Doctoral thesis conducted in the Biomedicine Doctorate Program of the University of Granada, Spain.

Additional Information and Declarations

Competing Interests

Author Contributions

Human Ethics

Clinical Trial Ethics

Data Availability

Clinical Trial Registration

The authors declare that they have no competing interests.

Rodrigo Lizama-Pérez conceived and designed the experiments, performed the experiments, analyzed the data, prepared figures and/or tables, authored or reviewed drafts of the article, and approved the final draft.

Luis Javier Chirosa-Ríos conceived and designed the experiments, performed the experiments, analyzed the data, prepared figures and/or tables, authored or reviewed drafts of the article, and approved the final draft.

Guido Contreras-Díaz conceived and designed the experiments, performed the experiments, analyzed the data, prepared figures and/or tables, authored or reviewed drafts of the article, and approved the final draft.

Daniel Jerez-Mayorga conceived and designed the experiments, performed the experiments, analyzed the data, prepared figures and/or tables, authored or reviewed drafts of the article, and approved the final draft.

Daniel Jiménez-Lupión conceived and designed the experiments, performed the experiments, analyzed the data, prepared figures and/or tables, authored or reviewed drafts of the article, and approved the final draft.

Ignacio Jesús Chirosa-Ríos conceived and designed the experiments, performed the experiments, analyzed the data, prepared figures and/or tables, authored or reviewed drafts of the article, and approved the final draft.

The following information was supplied relating to ethical approvals (i.e., approving body and any reference numbers):

The study protocol was approved by the Scientific Ethics Committee of the University of Granada, Spain (2380/CEIH/2021) and of the Universidad San Sebastián, Chile (55-2021-20).

The following information was supplied relating to ethical approvals (i.e., approving body and any reference numbers):

The study was conducted at the Basic Sciences Laboratory, Universidad San Sebastian, Valdivia, Chile. The study protocol was approved by the Scientific Ethics Committee of the University of Granada, Spain (2380/CEIH/2021) and of the Universidad San Sebastián, Chile (55-2021-20).

The following information was supplied regarding data availability:

The raw measurements are available in the Supplemental Files.

The following information was supplied regarding Clinical Trial registration:

NCT05469191.

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
