# Peer review of "Effect of sit-to-stand-based training on muscle quality in sedentary adults: a randomized controlled trial"

_PeerJ, doi:10.7717/peerj.15665_

## Round 0.1 · original submission · Major Revisions

Dear Authors,

The reviewers and I have completed our evaluation of your manuscript and recommend a major revision before re-submission.

Please review the comments and resubmit your revised manuscript.

Reviewer 1 ·

Basic reporting

Please, see the pdf file with my comments.

Experimental design

Please, see the pdf file with my comments.

Validity of the findings

Please, see the pdf file with my comments.

Additional comments

Please, see the pdf file with my comments.

Annotated reviews are not available for download in order to protect the identity of reviewers who chose to remain anonymous.

Reviewer 2 ·

Basic reporting

This is a well-written paper and has an interesting approach. I hope the comments improve the final manuscript and help the research community in the field to understand the problems of this population. However, there are some considerations to be made and some questions that would assist, especially, in the methods sections, data analysis, and presenting the results in this manuscript. In addition, the rationale for comparing the two training volumes (5STS and 10STS) was not clear.
# Abstract
1. Standardize the acronyms throughout the text;
2. Line 23 – Change “Functional variables” to “Muscle function variables”.
3. Line 26 – add “by Ultrasonography”.
4. I think the information regarding the final sample data should be included in the results section;

Main text
#Introduction
The aim of this study was to investigate the effects of the two sit-to-stand (STS) modalities on functional performance and muscle architecture in the adult population.
The hypothesis of the study was that STS training would increase functional performance and produce adaptations in the architecture of the vastus lateralis muscle (VL) in adults. At no time do the authors address the comparison between 5STS and 10STS. I wonder if this comparison is relevant. Why compare the two training modalities? The authors must present a justification for carrying out the two training modalities. It wasn't clear!
Objective (lines 74-75)
1. Based on the International Classification of Functioning, Disability, and Health, the authors did not evaluate “functional performance”. I suggest “muscle function and architecture”
2. Add "sedentary adult population" just like in the title.

#Figures 2 - Please, check formatting.
#Table 1 - Check the p-value in the legend.
#Tables 1 and 2 - Present the 95%CI of the differences (%change).

Experimental design

1. Some topics from the CONSORT checklist were not found in the manuscript. They must be described.
- Completely defined prespecified primary and secondary outcome measures, including how and when they were assessed;
- Type of randomization; details of any restriction (such as blocking and block size);
- Mechanism used to implement the random allocation sequence (such as sequentially numbered containers), describing any steps taken to conceal the sequence until interventions were assigned.
- For each group, the numbers of participants who were randomly assigned, received intended treatment, and were analyzed for the primary outcome
- Dates defining the periods of recruitment and follow-up
- A table showing baseline demographic and clinical characteristics for each group

2. How the participants were recruited? Is this a convenience sample?
3. Line 98 – “Only”?
4. Randomization - The authors need to clarify the mechanism used to implement the random allocation sequence. Were participants randomly allocated at the same time?
5. Is not a blinded study. This must be cited as a limitation of the study.
6. Sit-to-stand test - Was used verbal stimuli? Please, include a reference for STS.
7. In my opinion the functional assessment performed does not include the most popular variables from SST: the time (in seconds) required to perform 5 and 10 consecutive repetitions of sitting down on and getting up from a chair. Could the authors insert this data?
8. What was the sequence of the tests (STS)? Please clarify!

Validity of the findings

1. Statistical Analysis
This study is a prospective study with three independent groups (control vs. interventions) evaluated in two moments (8-week follow-up). First, I strongly suggest to the authors perform data analysis by the "intention-to-treat" principle.
Second, the comparisons between differences (% changes) can be performed by the one-way ANOVA, for variables with a statistical significance to the within-group differences. They are very interesting ancillary analyses and could be shown.

2. The baseline demographic and anthropometric characteristics for each group must be shown in a table.

Additional comments

No comment.

---

## Round 0.2 · accepted · Accept

Your manuscript has been accepted for publication. Congratulations!

Reviewer 1 ·

Basic reporting

No comment

Experimental design

No comment

Validity of the findings

No comment

Additional comments

The authors did a great job in reviewing the manuscript as requested. I have no further comments. Congratulations on a very interesting study. All the best.